# Genoprotective Effect of Some Flavonoids against Genotoxic Damage Induced by X-rays In Vivo: Relationship between Structure and Activity

**DOI:** 10.3390/antiox11010094

**Published:** 2021-12-30

**Authors:** Miguel Alcaraz, Amparo Olivares, Daniel Gyingiri Achel, José Antonio García-Gamuz, Julián Castillo, Miguel Alcaraz-Saura

**Affiliations:** 1Radiology and Physical Medicine Department, School of Medicine, University of Murcia, 30100 Murcia, Spain; amparo.o.r@um.es (A.O.); gamuz@um.es (J.A.G.-G.); Miguel.Alcaraz@um.es (M.A.-S.); 2Applied Radiation Biology Centre, Radiological and Medical Sciences Research Institute, Ghana Atomic Energy Commission, Legon, Accra GE-257-0465, Ghana; d.gachel@gaecgh.org; 3R&D Department, Iff Murcia Natural Ingredients, Site Plant: Nutrafur, Camino Viejo de Pliego, Km. 2, Box 182, 30820 Alcantarilla, Spain; jcsanchez@ucam.edu

**Keywords:** radiation effects, radioprotectors, micronucleus, flavonoids

## Abstract

Flavonoids constitute a group of polyphenolic compounds characterized by a common gamma-benzo- pyrone structure considered in numerous biological systems to possess antioxidant capacity. Among the different applications of flavonoids, its genoprotective capacity against damage induced by ionizing radiation stands out, which has been related to antioxidant activity and its chemical structure. In this study, we determined the frequency of appearance of micronucleus in vivo by means of the micronucleus assay. This was conducted in mice treated with different flavonoids before and after exposure to 470 mGy X-rays; thereafter, their bone marrow polychromatophilic erythrocytes were evaluated to establish the structural factors enhancing the observed genoprotective effect. Our results in vivo show that the presence of a monomeric flavan-3-ol type structure, with absence of carbonyl group in position C4 of ring C, absence of conjugation between the carbons bearing the C2 = C3 double bond and the said ring, presence of a catechol group in ring B and characteristic hydroxylation in positions 5 and 7 of ring A are the structural characteristics that determine the highest degree of genoprotection. Additionally, a certain degree of polymerization of this flavonoid monomer, but maintaining significant levels of monomers and dimers, contributes to increasing the degree of genoprotection in the animals studied at both times of their administration (before and after exposure to X-rays).

## 1. Introduction

Flavonoids are a widely extended group of polyphenolic compounds characterized by having a common benzo-γ-pyrone structure that has been associated with antioxidant capacity in different biological systems [1,2,3,4]. More than 8000 compounds with flavonoid structure have been identified, since there are numerous combinations with multiple substituents of hydroxyl, methoxyl, O- and C-glycoside groups on the basic benzo-γ-pyrone structure (C6-C3-C6) [4,5,6,7]. Flavonoid synthesis seems to be associated with lignification processes in leaves and logs and with ultraviolet light processes in flowers [8,9], as well as with different types of stress, especially induced through fungal and microbial infections, currently being considered as part of the immune system of plants [10,11,12,13,14].

These compounds not only play important physiological and ecological roles in plants, they also possess important commercial value due to their multiple applications in the agro-food and pharmaceutical industries [4,6,15,16,17,18,19,20,21,22]. Numerous studies have demonstrated that these compounds have a wide range of medical applications, for example, as antibacterial [23,24], anti-inflammatory [25], antioxidant [26], antineoplastic [27] and antigenotoxic [28] agents. Flavonoids can act by protecting DNA against oxidative damage, inactivating carcinogens, inhibiting the expression of genes and enzymes responsible for the activation of procarcinogenic substances and activating systems responsible for xenobiotic detoxification [5,29]. Flavonoids have shown the ability to modify the activity of enzymatic systems in mammals in vitro (Kinases, phospholipases, ATPases, lipooxygenases, cyclooxygenases, phosphodiesterases, etc.) [4,5,30,31]. Some studies have related these effects to the chemical structure of flavonoids, describing that their antioxidant activities, enzyme inhibition or antineoplastic capacities depend on small variations in their basic chemical structure [5,30,31,32,33,34,35,36,37,38,39].

In relation to ionizing radiation, different studies have established a relationship between the radioprotective and antimutagenic capacities of some flavonoids with their chemical structures and antioxidant capacities [40]. On the other hand, other flavonoids have been shown to have mutagenic, genotoxic, and even radiosensitizing capabilities [23,41,42]. Therefore, although flavonoids have a common basic chemical structure, there must be important structural factors that can condition the radioprotective activity, such as the degree of oxidation of the structure (flavanone, flavone), the substituents (position, number and nature of the groups in the A and B rings of the flavonoid skeleton and the presence of glycosylation [4,5,34,43].

In this study, we intend to relate the chemical structures of different flavonoids with their in vivo antigenotoxic capacities determined by the rodent bone marrow micronucleus technique.

## 2. Materials and Methods

### 2.1. Chemicals and Reagents

Diosmin, quercetin, rutin and rosmarinic acid were obtained from Extrasynthese (Genay, France); vitamin C and 6-n-propyl-2-thiouracil-6c (PTU) were obtained from Sigma-Aldrich (Madrid, Spain); dimethyl sulphoxide (DMSO) was obtained from Merck (Darmstadt, Germany); amifostine (WR-2721, Ethyol^®^) was obtained from Schering-Plough S.A., (Madrid, Spain); and Zoledronic acid (Zometa^®^) was obtained from Novartis Farmaceutica (Barcelona, Spain). Apigenin, carnosic acid, grape procyanidins short (P short), medium (P medium) and long (P long) (according to their degree of polymerization), grape seed extract (all degrees of polymerization), olive leaf extract, citrus extract and green tea extract were obtained from Nutrafur (Alcantarilla, Spain). Additional information on the composition and characteristics of the extracts used is shown in the Appendix A. Pycnanthus angolensis seed extract (PASE) were obtained as previously described [44].

Phosphate buffered saline (PBS), methanol, and sodium bicarbonate were obtained from Sigma-Aldrich Chemicals S.A. (Madrid, Spain), and fetal bovine serum was obtained from Gibco (Life Technologies S.A., Madrid, Spain).

### 2.2. Micronucleus Assay in Mouse Bone Marrow (Micronuclei in Polychromatic Erythrocytes (MnPCEs))

Eleven-week-old male Swiss mice weighing 26–32 g and distributed in groups of 4 animals were used for each of the groups tested. The animals were kept at the Animal Service Laboratory of the University of Murcia (REGAES300305440012), and all procedures and techniques were approved by the Ethical Committee of the Autonomous Community of the Region of Murcia (Spain) (CECA: 510/2019).

The in vivo micronucleus test was conducted on mouse bone marrow, as previously described [45]. Briefly, 24 h after X-ray exposure, the mice were sacrificed and the two femurs from each mouse were extracted, the proximal and distal epiphyses were cut and their medullary canal were gently flushed and washed with calf serum to obtain the bone marrow cells. These cells were dispersed by gentle and repeated pipetting and collected by centrifugation at 1000 rpm for 5 min at 4 °C. The pellets obtained were resuspended in a small volume fetal calf serum (0.2 mL) and dispersed on cold slides to obtain bone marrow smears (4 slides per mouse). After 24 h of air drying at room temperature, the smears were stained with May-Grünwald/Giemsa [45]. With this method, polychromatic erythrocytes (PCE) stain reddish blue and normochromatic erythrocytes (NCE) stain orange, while nuclear material stains dark purple (Figure 1). The slides were examined at 400× magnification using a Zeiss light microscope (Zeiss, Oberkochen, Germany). Subsequently, the preparations were digitized using a Leica SCN400F scanner combined with Existing Digital Image Hub (version 3.0, Leica Microsytems, Buffalo Grove, IL, USA). The digitized preparations were evaluated by three specialists in a double-blind study to determine the number of micronuclei in 4000 polychromatic erythrocytes (PCEs) per mouse. The results were expressed in MnPCEs/1000 PCEs, figures were rounded to obtain only whole numbers. To ensure that the substances tested were nontoxic, the number of normochromatic erythrocytes, polychromatic erythrocytes, and total erythrocytes, as well as the rate of appearance between them in each animal, were also determined.

### 2.3. Irradiation

The experimental animals were irradiated in an Andrex SMART 200E X-ray generator (Yxlon International, Hamburg, Germany) with the following characteristics: 120 kV, 1.5 mA, 2.5 mm Al filtration and dose rate of 0.653 mGy/s at a focus-to-object distance (FOD) of 35 cm. The animals were subjected to whole-body irradiation at room temperature and kept conscious but immobilized in a plastic structure designed for the occasion. The total exposure time to ionizing radiation was 12 min for a total dose of 470 mGy. The structure was rotated 90° every three (3) minutes to compensate for the anodic effect of the X-ray tube. The radiation dose administered was monitored at all times by means of a radiation dosimeter located inside the X-ray cabin next to the animals (UNIDOS^®^ Universal Dosimeter with PTW Farme^®^ ionization chambers TW30010 (PTW-Freiburg, Freiburg, Germany)). The final radiation dose was confirmed by means of thermo-luminescent dosimeters (TLDs) (GR-200^®^; Conqueror Electronics Technology Co. Ltd., Shenzhen, China).

To construct the dose–response curve, the frequency of appearance of micronuclei in mouse bone marrow polychromatic erythrocytes (MnPCEs/1000) was determined after exposure to 28 different X-ray doses between 0 and 800 mGy, each corresponding to each of the X-ray doses received by untreated (control) animals. Each experimental cohort consisted of 4 animals; these were treated with test substances prior to and/or after exposure to X-rays. Test substances were administered intraperitoneally at a dose of 300 mg/ml dissolved in saline (grape seed extract, P medium, P short, P long, rosmarinic acid, citrus extract, vitamin C and PTU) or in 5% DMSO (carnosic acid, rutin, apigenin, olive leaf extract, disomin, PASE, green tea extract, quercetin and DMSO). Amifostine (Ethyol^®^) was used directly as prepared from the (commercial) source. A 0.3 mL volume of the solution was administered to each animal by intra-abdominal injection in the left upper quadrant. In the cohort that received test substances before exposure to X-rays the substances were administered 60 min before irradiation. In the groups treated after exposure to X-rays, the substances were administered immediately at the end of the exposure (within 5 min after X-ray exposure).

### 2.4. Statistical Analysis

The increase in the frequency of the appearance of micronuclei in PCEs was analyzed as an expression of the genotoxic effect induced by X rays. Likewise, the reduction of the frequency of micronuclei in PCEs treated with the substances and irradiated showed the genoprotective capacity of each substance. The statistical analysis consisted of the use of analysis of variance to compare the mean differences across groups. Likewise, linear and polynomial regression and correlation analyzes were also conducted. Statistically significant results have been considered when *p* is less than 0.05 (*p* < 0.05)

In addition to these, two other parameters, namely Magnitude of Protection (MP) and Dose Reducing Factor (DRF), were also determined. The MP is the protection capacity of a radioprotective substance, expressed as a percentage of protection [46], using the formula:
MP (%) = (F_MNcontrol irradiated_ − F_MN treated irradiated_/F_MNcontrol irradiated_) × 100
where F_MNcontrol irradiated_ is the frequency of MnPCEs/1000 PCEs in the irradiated control group, and F_MNtreated_ is the frequency of MnPCEs/1000 PCEs in the group treated with test substances and irradiated.

The DRF is the quotient between the radiation dose necessary to produce a given effect in the presence of a radioprotective compound (MnPCEs/1000 PCEs in the group treated with the test substances and irradiated) and the radiation dose necessary to produce the same effect in the absence of said radioprotective compound, which makes it possible to determine the reducing/radioprotective capacity of the tested substance [47]. It is expressed by the following formula:DRF = DFMNtreated/DFMN
where DFMNtreated is 470 mGy (which is the radiation dose produced by a given frequency MnPCEs/1000 PCEs in the group treated with the radioprotective substances and irradiated with 470 mGy), and DFMN is the radiation dose necessary to produce the same frequency of MnPCEs/1000 PCEs in irradiated animals using the dose–response curve of this study (D (mGy) = −63.44 + 24.88 · y; see Figure 2).

## 3. Results

The dose–response curve obtained after total body irradiation showed a linear relationship and a 96.07% (R^2^) reliability within the dose range of 0.5 mGy to 800 mGy. Conditions used for this study permitted the estimation of radiation dose as a function of the number of micronuclei (MN) formed in the bone marrow polychromatic erythrocytes (Figure 2). To estimate the DRF, the radiation dose necessary to produce a certain number of micronuclei was obtained from the equation
D = −63.44 + 24.88 · y
where y is the number of MNs obtained at 470 mGy in the presence of the radioprotective substance tested and D is the X-ray dose (in mGy) necessary to obtain the same number of micronuclei in untreated animals.

The intraperitoneal administration of the tested substances did not present statistically significant differences in the frequency of appearance of MN with respect to the untreated control animals; expressing an absence of genotoxic effect produced by the administration of said test substances. However, when zoledronic acid (Zometa^®^) was administered, a significant increase in the frequency of MN was determined compared to control animals, reaching a count of 16 ± 2 MN/1000 PCEs, showing significant differences (*p* < 0.001) in comparison with controls, and that expresses a genotoxic capacity of zoledronic acid.

Exposure of the untreated control animals to 470 mGy X-rays produced a significant increase in the frequency of MN, reaching a count of 19 ± 2 MN/1000PCEs. This represents a significant difference in the frequency of the appearance of MN with respect to non-irradiated control animals (*p* < 0.001) and expresses a chromosomal and genotoxic damage induced by ionizing radiation (Figure 3).

### 3.1. Administration of Test Substances Prior to Exposure to Ionizing Radiation

Exposure of the animals treated with the different test substances to 470 mGy of X-rays before exposure to radiation produced different effects on the frequency of appearance of MN/1000 PCEs, expressing different degrees of genoprotection (Figure 3). In the first five test substances represented in Figure 3 (see numbers 1–5), a significant reduction in the frequency of MN/1000 PCEs was observed in the treated and irradiated animals (*p* > 0.001), expressing a significant protection capacity against the chromosomal damage induced by ionizing radiation. The next 12 substances administered before radiation exposure (see numbers 6–17, Figure 3) also showed significant reduction in the frequency of MN/1000 PCEs compared to irradiated control animals (*p* < 0.01), which also expresses a significant but less intense degree of genoprotection compared to the previous substances. The sulfur substances (amifostine, DMSO and PTU) were among the substances that presented the lowest genoprotective capacity (*p* < 0.05). Finally, the absence of genoprotection in animals treated with quercetin is very obvious. Similarly, the genosensitizing effect of zoledronic acid (*p* < 0.01) is expressed an increase in chromosomal damage induced by X-rays, which can be interpreted as a radiosensitizing effect of these substances (Figure 3).

To address the problem of low solubility in aqueous medium, some of the test substances used that were lipid in nature or of low polarity were dissolved in DMSO; for this reason, 5% DMSO was tested in saline serum.

Figure 4 shows the amount of protection conferred by test substances administered prior to exposure to X-ray. Of all the substances analyzed, grape seed extracts and the medium chain procyanidins (medium degree of polymerization, P medium) exhibited the highest genoprotective capacity with a magnitude of protection of 68%. In this comparative study of radioprotective capacities, it is worthy of note that amifostine, the only radioprotective substance approved by the FDA for clinical use, expressed a magnitude of protection of 37%.

Figure 5 shows the dose reduction factors (DRF) obtained when the different substances were administered prior to exposure to 470 mGy X-rays. Using the dose–response equation obtained in our study, we determined the radiation doses (in mGy) that would cause MN formation when mice were administered the test substances prior to radiation exposure. The DRF of value 5.8 obtained with medium chain procyanidins (P medium) and grape seed extracts deserves to be highlighted. This value is much higher than that obtained for amifostine (1.7), and that determined for vitamin C, which presented a DRF of 2.4.

When the test substances used were administered before exposure to ionizing radiation, the medium chain procyanidins and grape seed extracts elaborated the highest genoprotective capacity of all the substances tested, presenting a micronucleus yield of 6 ± 2 MN/1000 PCEs after exposure to 470 mGy X-rays, a MP of 68% and a DRF of 5.8.

### 3.2. Administration of Test Substances after Exposure to Ionizing Radiation

When test substances were administered immediately after exposure to ionizing radiation, an overall decrease in the frequency of MN occurrence was observed relative to administration before irradiation exposure. Since this response is significantly lower for each of the test substances, it expresses a lower degree of genoprotection against chromosomal damage induced by ionizing radiation. Sulfur substances, such as amifostine, DMSO and PTU, presented a micronucleus frequency similar to that determined in irradiated control animals (Figure 6) and expresses a loss of their genoprotective capacity when administered after exposure to ionizing radiation. Additionally, the increased frequency of MN/1000 PCEs determined in animals treated with rosmarinic acid and the loss of the genoprotective capacity of rutin stand out.

As a consequence of the increase in MN/1000 PCEs previously determined, a lower magnitude of protection under these conditions was also observed, highlighting the loss of genoprotective capacities of amifostine (0%), DMSO (0%), PTU (0%), and rutin (0%), and a decrease was also observed for rosmarinic acid (16%) (Figure 7).

Figure 8 shows the DRFs of substances administered immediately after exposure to ionizing radiation. The values show that DRFs are much lower than what was determined when these substances are present in the biological environment prior to exposure to ionizing radiation. Under these conditions, procyanidins, grape seed extracts, and citrus extract are the substances that present the highest levels of dose reduction, in the following order: P short > citrus extract > grape seed extracts = P medium = carnosic acid.

Some of the substances and extracts used were insoluble in water, so they were dissolved in 5%DMSO to enhance their bioavailability. At the concentration of DMSO tested, no toxicity was determined in the animals, so we used this concentration of DMSO in our study to dissolve fat-soluble substances (carnosic acid, rutin, apigenin, olive leaf extract, diosmin, PASE, green tea extract, quercetin and DMSO). For this reason, DMSO, as an antioxidant and sulfur-containing substance, was included in our study protocol to assess its genoprotective capacity. DMSO only showed significant genoprotective capacity when found in the biological environment before exposure to ionizing radiation, and has no effect when administered after exposure to ionizing radiation. Therefore, it is possible to conjecture that in the fat-soluble substances dissolved in DMSO an additive or synergistic effect may occur between the genoprotective capacity of DMSO and the genoprotective capacity of the fat-soluble substance tested. Figure 9 shows the magnitudes of protection (MP) of the fat-soluble substances dissolved in DMSO after the eliminating for the genoprotective effect induced by DMSO in the animals when administered both before and immediately after exposure to ionizing radiation.

Table 1 shows the structural characteristics of the tested substances in relation to their Magnitudes of Protection (%), as well as the modifications due to possible additive effects of the substances dissolved in DMSO.

## 4. Discussion

The micronucleus assay (MN), which has been in use for decades, has become one of the most widely used methods for evaluating chromosomal damage caused by different chemical and physical factors [48,49]. Rodent bone marrow MN test is the most widely used micronucleus test for the determination of genotoxicity in vivo. It has been considered the most reliable test to evaluate the induction of chromosomal aberrations that are considered one of the main endpoints of mutagenicity. It is used in the identification of toxic agents and in the assessment of the risk of exposure to chemical substances. It has even been rated to be of greater importance for the assessment of these potential risks than numerous other assays, including the in vitro mammalian chromosome aberration assay [50]. Among the advantages of the in vivo micronucleus technique is the simple and easy-to-identify end point: the micronucleus is a component of DNA that remains in the cell cytoplasm when the main nucleus has already been expelled during erythropoiesis in mammals thus appear as a simple and easy-to-recognize structure inside the cell [45,50]. In bone marrow erythrocytes, MN is observed as a small, rounded body that contains chromatin that is visible in the cytoplasm, as it has staining characteristics similar to that of a cell nucleus [48]. Immature erythrocytes in bone marrow (polychromatic erythrocytes, PCEs) account for 50% of all red blood cells [48]. These have a bluish coloration that allows for easy differentiation from mature erythrocytes (normochromatic erythrocytes, NCEs) that are observed as a pink-orange cabbage in bone marrow smears stained with May-Grünwald-Giemsa [45] (Figure 1). MN is considered to be caused by DNA damage or genomic instability [48]. Although MN can occur as a result of natural processes, such as alterations in metabolism or aging, it can also be induced by many environmental factors, dangerous habits, and different diseases [48]. In this study, we used the PCE assay to evaluate the genotoxic effects of X rays by determining the frequency of appearance of micronuclei in PCEs. Similarly, a reduction in the frequency of appearance of these micronuclei after exposure to ionizing radiation made it possible to evaluate the genoprotective capacities of some substances tested which gave an expression of a decrease in chromosomal damage induced by ionizing radiation [49,51].

In our study, we established a linear relationship between the low doses of X-rays used and the frequency of appearance of micronuclei in PCEs from mouse bone marrow. We did not find any previous references in this regard. However, different studies have also described a similar relationship using other in vitro micronucleus techniques such as that of irradiated human lymphocytes by blocking cytokinesis with cytochalasin-B (Cyt-B) [51,52,53,54,55,56,57,58]. Other studies have occasionally described a linear-quadratic relationship [59,60,61,62]. These differences in in vitro studies have been attributed to the type of ionizing radiation used, the differences in the biological effectiveness of the radiation used [63], and the cell culture time [54]. Sometimes, these differences can also be explained by the technical modifications used, and even by the criteria for identifying the MNs used in the different laboratories [62].

The radiation doses used in these in vivo micronucleus studies are often very high, and increases in the frequency of appearance of MNPCEs are observed at 0.25 Gy [64,65], 0.5 Gy [64,66], 1 Gy [46,67], 2 Gy [64,68,69,70], 6 Gy [68], and even 8–10 Gy [65]. These studies show a significant increase in micronuclei yield in rodent bone marrow both for X-ray [66,68,70,71,72,73,74], gamma radiation [65], and even exposure to magnetic fields [74]. The spontaneous or basal frequency of micronuclei is always less than 5 MnPCES/1000 PCEs in the untreated control animals, which has been conserved as a quality parameter in the technique used [45,46,64,65,66].

We selected amifostine as a reference substance with radioprotective capacity, since it is a sulfur substance with sulfhydryl bonds (SH) and is the first chemical compound to be described with radioprotective capacity [46,68,69]. Although its clinical use is limited to the protection of the salivary glands during irradiation of patients with head and neck cancers and for the reduction of the side effects of chemotherapy, it is the only radioprotective substance used in radiation oncology [75,76,77]. The precise mechanisms by which amifostine and other sulfur substances exert their radioprotective actions have not been completely clarified. It has been suggested that at the molecular level, it is due to its ability to scavenge free radicals, its ability to donate hydrogen, and its ability to bind to biological targets, forming disulfide compounds; at the physiological level, it is by inducing hypoxia, hypothermia or shock; and at the organic level, by stimulating the recovery of cell populations [68,69,70]. At the structural level, it has been proposed that substances with radioprotective capacity that contain sulfur are characterized by possessing the following five chemical characteristics: presence of a latent sulfhydryl group protected by a phosphate; presence of at least one amino group separated from the sulfur atom by two or three carbon atoms; presence of a second amino group separated from the first by three carbon atoms; high solubility in distilled water at pH 6, which is indicative of a high pK value; and in addition, possessing an effective mean dose of 600 mg/kg, which is capable of reducing the toxicity properties of this type of substances [78].

In our study, amifostine showed a moderate genoprotective capacity in vivo, behaving like a classic radioprotector that only presents protection when found in the biological environment before exposure to ionizing radiation, losing its radioprotective activity when administered immediately after exposure to radiation [47]. Kopjar et al. (2006) [79] describe a genoprotective effect of amifostine administered before exposure to irradiation. Furthermore, Müller et al. (2004) [80], also determined its genoprotective capacity under these conditions using the Comet assay. On the other hand, DMSO is another sulfur-containing antioxidant substance considered to be a classic radical scavenger with a high capacity to capture hydroxyl radicals in vitro [81,82]. In this study, DMSO and PTU were found to have a lower genoprotective capacity than amifostine, which could be explained by the absence of some of the chemical structural properties previously described for evaluating the radioprotective capacities of these sulfurized substances [78]. Unfortunately, the effective radioprotective doses of many of these types of substances are usually toxic and only have protective capacity if they are found inside the body or the biological environment before exposure to radiation [68,70,78,81].

We also used vitamin C and zoledronic acid for quality control of our results in this study. Our results confirm the in vivo genoprotective capacity of vitamin C administered before and after exposure to ionizing radiation, which has been previously described with different in vivo [71,73] and in vitro micronucleus techniques [40,44,51,83]. On the other hand, our in vivo results confirm the increase in genotoxic damage induced by zoledronic acid previously described with in vitro micronucleus techniques, showing an additive and/or synergistic effect of zoledronic acid and ionizing radiation that has been interpreted as a genosensitizing effect at both times of administration (before and after exposure to ionizing radiation) [84].

Our results show that there is no single chemical structure that is the protagonist of genoprotection, although a proportional relationship with the antioxidant capacities of the test substances was observed. In this study, we tested several flavonoids that have shown very different responses, while some show intense genoprotection (procyanidins), others show a significant degree of radiosensitivity (quercetin). In our study, flavonoids that portrayed the highest genoprotective capacity were the procyanidins (P short, P medium, P long) and grape seed extract extracted from grapes. Analyzing the genoprotective response of the different flavonoids tested, in general, it can be said that in order to consider a substance as being radioprotective, it must meet two conditions in its chemical structure: first, a molecular structure capable of scavenging free radicals; and, secondly, that the compound formed should be as stable as possible to avoid the propagation of the harmful effects induced by ionizing radiation.

The antioxidant capacity of flavonoids can be explained according to their chemical structure, which can determine their genoprotective capacity, based on the structural elements described: catechol groups, conjugated rings with carboxyl groups, carbonyl groups with α, β unsaturated bonds etc. [4,23,83]. Thanks to the relative structural similarity of these flavonoids, it is possible to make a global hypothesis about the influence of these structural elements on the genoprotective activity of the different flavonoids studied and to interpret the results obtained. Following the schemes proposed by different authors, the different points at which flavonoids could exert their antioxidant and free radical scavenger activity would be the following [4]:Antiradical activity (●OH, hydroxyl.Antiradical activity (O_2_^●−^, superoxide).Metal chelating activity.Antilipoperoxidant activity (R●, alkyl; ROO●peroxy; RO●, alkoxy).The activation and/or inhibition of various enzymes related to this oxidative metabolism.

During an exposure to low LET ionizing radiation, approximately 65% of DNA damage is caused by the indirect effect of free radicals such as ●OH that are produced from the radiolysis of surrounding water molecules, even in the absence of oxygen [44]. An important function of flavonoids and other polyphenolic compounds is the elimination of these free radicals. Their genoprotective effect on cells is attributed to the inhibition of ROS before they can interact with DNA and other macromolecules. Since these ROS are mostly very short-lived, they have to interact with DNA immediately after their ionizing radiation-induced production [23,83]. Flavonoids can donate a hydrogen atom or electron from their hydroxy groups to free radicals, resulting in free radical repair and the formation of an inert molecule. The phenoxyl radicals produced are stabilized by delocalization of the unpaired electron within the aromatic structure of the flavonoid. The stabilization of the phenoxyl radical induced by ROS has been attributed to the o-dihydroxy group on the B ring and 3-hydroxy and 4-OXO groups and C2 = C3 double bond on the benzopyran ring [23]. Consequently, these flavonoids are excellent free radical scavengers due to the high reactivity of their hydroxyl substituents and the resonance effects of their unique polyphenyl [85]. High in vitro antioxidant activities of flavonoids have been considered to be related to their genoprotective effects. Thus, grape seed extracts, which contain high levels of polyphenolic compounds, have a greater capacity to scavenge ABTS radicals (2, 2′-azino-bis (3-ethylbenzothiazoline-6-sulfonic acid), diammonium salt, and are more genoprotective than rutin, DMSO or vitamin C [23,71].

The quenching capacity of flavonoids against singlet oxygen fundamentally depends on the conjugation of ring B with the 4-oxo function of ring C through a C2 = C3 double bond, the presence of a hydroxy group in position 3 that enhances said effect also needs highlighting [23]. For superoxide anion uptake, the negative influence of the catechol-containing B ring structure has been implicated. This is believed to be related to the generation of hydrogen peroxide, a potent oxidizing agent which contributes to the propagation of cellular oxidation reactions and may even permit reverse action, i.e., a pro-oxidant activity [23]. This negative influence that would cause the generation of hydrogen peroxide in the medium could be carried out according to the following reaction [4]:Flav-B-ringdiOH + O_2_^●−^·Flav-B-ringOO^●^^−^ + H_2_O_2_

The complexation of metal cations such as Fe^2+^/Fe^3+^ and Cu^+/^Cu^2+^, responsible for the generation of ●OH radicals, through the Fenton mechanism, requires the presence of at least two adjacent hydroxy groups, either in ring A or B, or an interaction between the 4-oxo group and the hydroxy groups in position 3 and/or 5 [4].

In relation to metal chelating activity, it has been demonstrated that flavonoids can cross the plasma membrane to chelate iron and eliminate intercellular redox activity. This removal of iron inhibits the oxidative process induced by H_2_O_2_ in which iron participates [86], thereby decreasing oxidative stress-induced DNA damage produced by iron [23,86]. However, a paradoxical effect has also been described in some flavonoids. Although flavonoids generally show antioxidant activity, they can change to pro-oxidants in the presence of transition metals [23,86]. Thus, although flavonoids react directly with free radicals as antioxidants, their chelation of metal ions could lead to the production of ROS. Flavonoids can bind to transition metal ions such as Cu^2+^ and Fe^3+^; metals with the highest redox activity in living cells. All types of flavonoids have been considered to have three domains capable of reacting with metal ions: the 3′, 4′-dihydroxy system located on ring B, the 3-hydroxy or 5-hydroxy groups on ring C, and the 4-carbonyl group in ring C [87]. The reduction of Cu^2+^ and Fe^3 +^ produced by phenolic and polyphenolic compounds can form superoxide radical anions by a reduction of a single electron from the oxygen molecule [88]. The superoxide radical, in turn, is converted to hydrogen peroxide and a hydroxyl radical, leading to the formation of a DNA adducts [88]. Fundamentally, this pro-oxidation activity has been considered responsible for the ability of flavonoids to increase cellular toxicity through increased DNA damage [89]. Traditionally, copper and zinc are the main metals naturally associated with chromosomes, so the presence of Cu^2+^ instead of Fe^3+^ would be the most important factor increasing DNA damage produced by the pro-oxidation action of these flavonoids [90]. It is important to highlight the location of the metals with redox activity because they generate hydroxyl radicals and other ROS that have a very short lifespan and do not penetrate deep into the cell medium. In order to damage DNA, these reactive substances must originate in the vicinity of the DNA. It has been established that both the copper-mediated antioxidant and pro-oxidant activities of polyphenolic compounds depend on the number and position of the hydroxy groups [89]. Quercetin has been reported to undergo more autoxidation and cause more DNA damage due to the reduction of Cu^2+^ to Cu ^+^ compared to other flavonoids (kaempferol and morin). Quercetin binds to Cu ^2+^ in close proximity to DNA to form a DNA-Cu^2+^-quercetin complex. Cu^+^ and superoxide anions are generated through the autoxidation of the quercetin-Cu^2+^ complex, which leads to the generation of hydroxyl and hydroperoxyl radicals that cause oxidation and can therefore produce additional damage to DNA. Similarly, it has been established that in the presence of quercetin or Cu^2+^ alone, there is no increase in these DNA lesions. [88]. Under these conditions, it has been proposed that the flavonoid-Cu^2+^-DNA complex is responsible for the carcinogenic and mutagenic effects of flavonoids due to its ability to increase oxidative stress and increasing DNA damage. All these observations were obtained from different isolated in vitro DNA studies, and thus it has been suggested that caution must not be thrown to the wind when extrapolating these results to humans. Our in vivo results show that quercetin does not have genoprotective capacity. On the contrary, we determined a genosensitizing activity, where it was found to increase the genotoxic effect of ionizing radiation, seemingly acting as a pro-oxidant substance. On the other hand, this also allowed us to postulate why rutin (the glycosylated form of quercetin) does not have these effects since its special planar arrangement would reduce the interaction with these metals in the vicinity of DNA [23].

The activation and/or inhibition of various enzymes related to this oxidative metabolism is more complex, since it acts at the level of specific receptors. In these cases, the presence and/or absence of simple hydroxy groups, the esterification of any of them, the planar or non-planar spatial structure in flavonoids, etc., may be determining factors. These elements can radically increase or decrease the activity of these compounds [37,38,39,40,71,83], but their assessment in this work was not possible. Administration of procyanidin-rich grape seed extracts (proanthocyanidins) has been reported to reduce radiation-induced oxidative stress in different mouse tissues, which are attributed to a significant increase in SOD, catalase, and GPx activity [91]. In general, it is postulated that the administration of flavonoids before exposure to ionizing radiation leads to an enhancement of enzymatic and non-enzymatic antioxidant status, indicating that they contribute to antioxidant capacity maintenance of the cell [23,92].

The results obtained in our work shows different protection capacities of the tested substances depending on the time of their administration with respect to X-ray exposure. This is possibly due to the fact that when the agents are administered after exposure to X-rays, the hydroxyl and superoxide radicals produced by the radiation have disappeared from the medium, and therefore, their action will be fundamentally on the cascade of reactions that they have produced. This situation implies a modification of the basic radiobiological concepts, since the harmful effect induced by ionizing radiation can be minimized even after the exposure [14]. In this context, the differentiation between anti-radical activities (vs. superoxide anion and hydroxyl radicals) and anti-lipoperoxidants (vs. lipoperoxy radicals) proposed by Pincemail et al. [93] seems reasonable and could explain the different behavior of the compounds tested in terms of their action against lipid peroxidation processes. Exposure to gamma radiation increases cellular oxidation processes, causing the successive formation of superoxide radicals (O_2_^●−^), hydrogen peroxide, and hydroxyl radicals (^●^OH) [47,94]. At this time, the massive generation of ^●^OH radicals is of special interest, since they are considered the most cytotoxic, and can possibly be mitigated by the administration of water-soluble genoprotective substances such as amifostine, PTU, DMSO or rosmarinic acid.

However, when the generation of these hydroxyl radicals is massive, they interact with cell phospholipoid structures, inducing lipid peroxidation processes and gradually producing lipoperoxy radicals (R^●^, RO^●^, RR^●^, ROO^●^, ROOR^●^). The continued accumulation of these lipoperoxy radicals could increase genotoxic effects in what could be considered a delayed reaction that would prolong the genotoxic effect for at least 24 h after exposure to radiation. This increase in lipid peroxidation leads to an increase in the activities of lipoxygenase, cyclooxygenase, and phospholipase, as well as increased secretion of lysosomal enzymes and arachidonic acid, thereby enhancing cellular inflammatory responses [47,94]. Substances such as carnosic acid, procyanidins, and vitamin C seem to act more effectively on these lipoperoxide radicals [46,71,83] (Figure 10, Table 2).

Antilipoperoxidant activity is not based on a single structural element, but depends on several combinations. In some cases, the determining element is the catechol structure in ring B, while in others, the conjugation of this ring with the 4-oxo function via the C2 = C3 double bond is the most important element [4,23]. However, in our study, the flavonoids with the highest antilipoperoxidant capacity were the procyanidins, which are devoid of C2 = C3 double bonds in the C-ring.

Glycosylated flavonoids have been found to have significantly decreased antioxidant capacities compared to non-glycosylated flavonoids. This decrease in antioxidant capacity could also lead to a decrease in its genoprotection capacity. Of the glycosylated flavonoids tested in this study, diosmin had the lowest genoprotective capacity. However, the data we have on quercetin and rutin show different results. In this study, rutin, a glycosylated quercetin [23,37], showed a significant genoprotective capacity that was not found in quercetin. The presence of the glycosylated structure in rutin seems to inhibit a putative pro-oxidant effect of quercetin, possibly preventing the chelation of metals in the vicinity of DNA [23], but it does not prevent it from showing a significant genoprotective effect in vivo (Table 2).

Different studies have shown that the degree of polymerization of flavonoids dictates their antioxidant capacities [4,37,51]. The higher the degree of polymerization, the greater their antioxidant capacity will be; therefore, they are expected to manifest a higher degree of genoprotection. However, these differences, which we revealed using earlier in vitro micronucleus techniques [51], were not so clearly established in this in vivo study (P medium > P short = P long).

In summary, the ability of flavonoids to capture free radicals and their in vitro antioxidant activities is determined by the presence of three main structural groups: (a) an ortho-diphenol group in ring B of its skeleton structure (catechol group), which confers greater stability to the aroxyl radical formed; (b) the double bond between carbons 2 and 3 of ring C, conjugated with the 4-oxo function; (c) hydroxy groups in positions 3 and 5 of rings C and A, respectively. Likewise, other structural elements, such as the presence of three adjacent hydroxy groups or the polymerization of the flavonoid structure participates in vitro, since this supports flavonoid conjugation of flavonoids. On the other hand, the mechanisms of action of these compounds are influenced by secondary structural factors, such as the presence or absence of sugars in some of their hydroxy groups (glycosylated forms), the total number of hydroxylated positions, and the existence or not of esterified hydroxyls, mainly with methyl groups. These structural elements of flavonoids are essential for the activation and/or inhibition of multiple enzyme systems involved in the metabolic cascade reactions, such as cyclooxygenases, lipoxygenases, phospholipases, prostaglandinsynthetases [4,5,23,37,38,39,40,41,42].

However, in our in vivo study, we determined that quercetin, which fulfils all previously described structural characteristics, does not show any genoprotective capacity, and even increases X-ray-induced genotoxic damage. In this study, we found that glycosylation does not abolish genoprotection in some flavonoids (diosmin) [95], and even when the chemical structures of some flavonoids known to lack genoprotective capacity (quercetin) are glycosylated (rutin), enhanced genoprotection against X-ray-induced DNA damage is displayed. Although rutin, a glycosylated flavonol, is a good genoprotectant when administered prior to irradiation, flavonols (such as quercetin and rutin) do not appear to be good in vivo genoprotectors. It seems that the presence of 4–5 free hydroxy groups, the presence of a C2 = C3 double bond conjugated with the C4 carbonyl carbon, and, above all, the fact that one of the hydroxy groups is located on position 3 of the C ring, introduces some level of instability to the product, which confers to it more pro-oxidant than antioxidant genoprotective properties. In any case, rutin itself has difficulty interacting with lipophilic radicals or intervening in the modulation of pro-inflammatory pathways by expressed as reduction in its genoprotective capacity when administered after exposure to ionizing radiation [37,39,90,96,97,98].

Additionally, flavones apigenin and diosmin are the group of flavonoids demonstrating the next highest genoprotective capacity. Neither of these have a catechol group in the B ring of their flavonoid skeleton, but both have a C2 = C3 double bond conjugated to a C4 carbonyl group. As in the case of citrus flavanones, it is possible that its genoprotective capacity is related, especially in the case of apigenin, to mechanisms other than the capture of free radicals, such as the ability to intervene in blocking certain pro-inflammatory metabolic processes [37,38,39,40,83,96,98,99].

The green tea catechins have a flavan-3-ol structure similar to that of procyanidins, but with two basic differences: the majority presence of galloyllated derivatives (the in vivo toxicity of gallic acid, adjacent tri-hydroxy groups are well known), and the total absence of polymerization. Both could explain the enormous difference in genoprotective capacity observed between tea and grape seed extracts [37,38].

Procyanidins, represented by the structure of a dimer in Figure 10, occupies the first position with respect to genoprotective capacity when administered both before and after exposure to irradiation. Their structural features can be summarized as follows: the presence of a monomeric flavan-3-ol type structure, with absence of carbonyl group at the C4 position of ring C, absence of conjugation between the carbons bearing the C2 = C3 double bond and the said ring, presence of a catechol group on ring B, characteristic hydroxylation at positions 5 and 7 of ring A. Additionally, there exists a certain degree of polymerization of this flavonoid monomer, while maintaining significant levels of monomers and dimers [40,44,51,71,74].

An assessment of other structural elements in monomeric structures, without polymerization, is complex, but it also suggests certain patterns of behavior. The next compound with the highest genoprotective efficacy is a citrus extract that is endowed with a very high content of glycosylated flavanones (naringin and, above all, neohesperidin). These compounds lack the catechol group on their B ring, as well as the C2 = C3 double bond. In addition, they possess neohesperidoside-like glycosylation in position 7 of ring A. In this case, its genoprotection capacity is more than likely related to mechanisms other than free radical uptake itself, some of which have already been mentioned [37,38,39].

The degree of polymerization of a flavonoid increases its antioxidant capacity by constituting a repetitive skeleton of that flavonoid in in vitro studies [37,51]. In our in vivo study, it was observed that monomers have a small genoprotective capacity, which increases to a maximum with a certain degree of polymerization (P short, P medium) and subsequently begins to decline with a significant increase in polymerization (P long). This could mean that large polymers do not interact well within the cell medium and/or with the radicals they must combat.

Finally, in this discussion, we have extensively and comparatively described the possible structural elements responsible for the genoprotective activity of the main polyphenolic compounds (flavonoids) evaluated in this study. However, we cannot fail to highlight the activity of three non-flavonoid compounds that nevertheless showed significant protective capacities when administered after irradiation: carnosic acid > olive extract = PASE. Looking for common active structural elements with the skeleton of a flavonoid, we would find the catechol-type structure (Phenol-ortho-dihydroxy) in all three cases, and the presence of a carboxylic acid group, already present in the case of carnosic acid and PASE, but directly generated in situ after administration in the case of olive extract (hydroxytyrosol + elenolic acid). Obviously, these elements are not enough to justify the significant activity observed, but a fundamental physical and chemical factor comes into play: the lipophilic character, defined by its lipophilic molecular structure (see Table 1), which allows it to interact directly with the liporadicals generated in the medium after irradiation.

We conducted a dose–response analysis of flavonoids whose genoprotection results differ from the results previously described in some studies. Clarification of these results would be useful to merit recognition of these substances for use as radioprotective agents in workers professionally exposed to ionizing radiation as well as for patients undergoing radiological procedures.

Given this apparent complexity, different studies have suggested the possibility of using mixtures of two or more genoprotective substances and/or the use of extracts made up of mixture of antioxidant substances, with the aim of looking for a possible additive or synergistic effect between the components or, failing that, obtaining a broad-spectrum radioprotector effect [100,101,102]. We used different types of extracts made up of antioxidant compounds that have shown significant genoprotective activities but do not show any significant additive effect when the substances are used individually. It is necessary to individually clarify the different mechanisms of action of each substance so that we can obtain mixtures with broad-spectrum genoprotection that are effective when administered before and/or after exposure to ionizing radiation [100,101,102].

## 5. Conclusions

Based on the in vivo results, the time of application of the tested flavonoid with respect to exposure to ionizing radiation modifies its genoprotective capacity, especially considering that cellular oxidative status is relatively different before and after irradiation. These differences in genoprotective capacities are associated with the basic skeleton or molecular structure of the flavonoids, and therefore will be the result of the “adaptive capacity” of the said skeleton to the “oxidative environment” existing at any given time. Obviously, it is complex to exhaustively identify the groups and flavonoid radical structures that are responsible for these genoprotective capacities. However, it seems possible to identify and establish some solid premises based on experimental in vivo data which can be extrapolated to these observed genoprotective capacities both before and after exposure to irradiation. These structural elements can be summarized as follows: presence of a monomeric flavan-3-ol type structure, with absence of carbonyl group at the position C4 of ring C, absence of conjugation between the carbons bearing the C2 = C3 double and the said ring, presence of a catechol group in ring B, characteristic hydroxylation at positions 5 and 7 of ring A; additionally, there exists a certain degree of polymerization of these flavonoid monomers, while retaining significant levels of monomers and dimers. These structural characteristics of procyanidins determined the highest degree of genoprotection in the animal studies, and applied to both times of their administration (before and after exposure to X-rays). Our studies provide important data for the structure–genoprotection relationships of flavonoids in vivo.

## Figures and Tables

**Figure 1 antioxidants-11-00094-f001:**
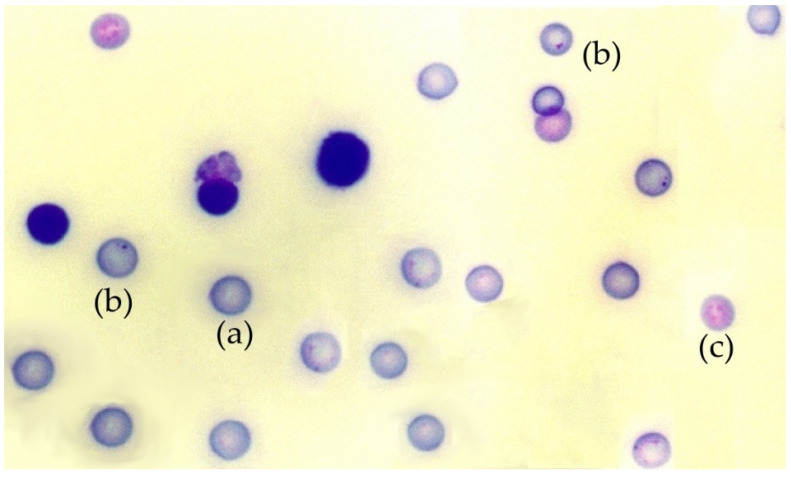
Photomicrograph of a mouse bone marrow smear stained with May-Grünwald-Giemsa: (**a**) polychromatic erythrocytes, (**b**) a micronucleus in polychromatic erythrocytes, (**c**) normochromatic erythrocytes.

**Figure 2 antioxidants-11-00094-f002:**
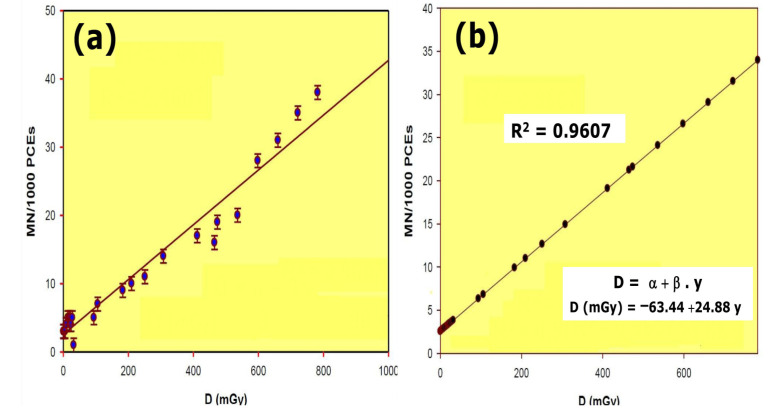
Dose–response relationship of radiation induced micronuclei in mouse: (**a**) results obtained; (**b**) linear regression and R squared value calculated from the results obtained.

**Figure 3 antioxidants-11-00094-f003:**
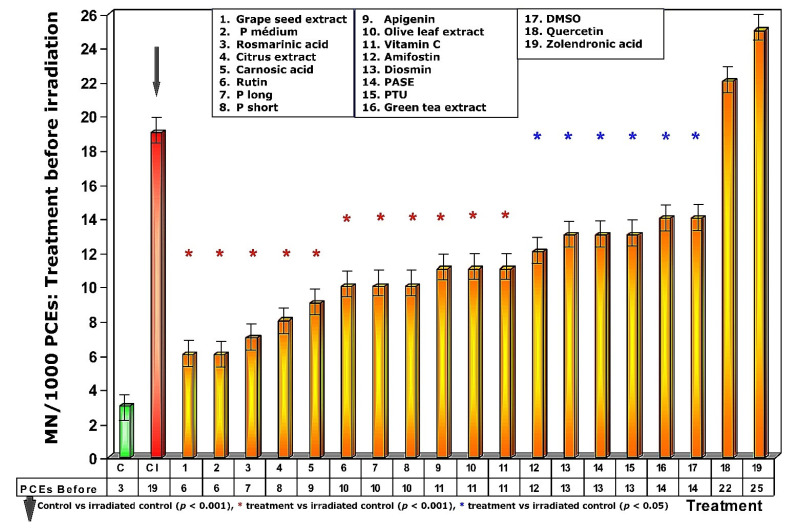
Frequency of micronuclei in polychromatic erythrocytes treated with test substances immediately before exposure to ionizing radiation.

**Figure 4 antioxidants-11-00094-f004:**
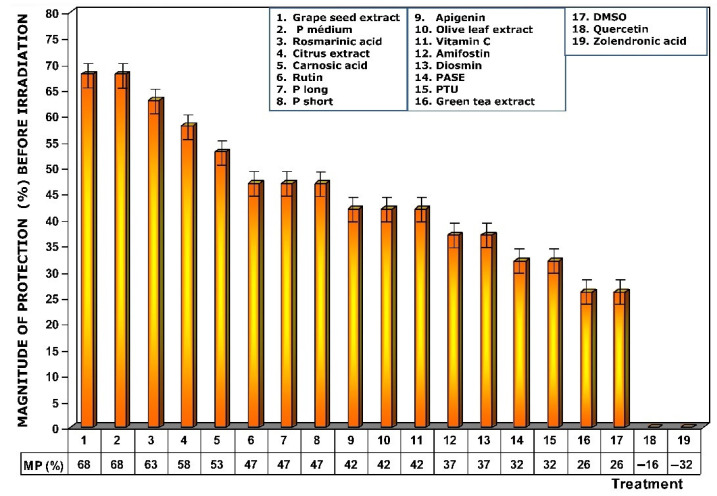
Magnitudes of protection obtained when tested substances were administered immediately before exposure to ionizing radiation.

**Figure 5 antioxidants-11-00094-f005:**
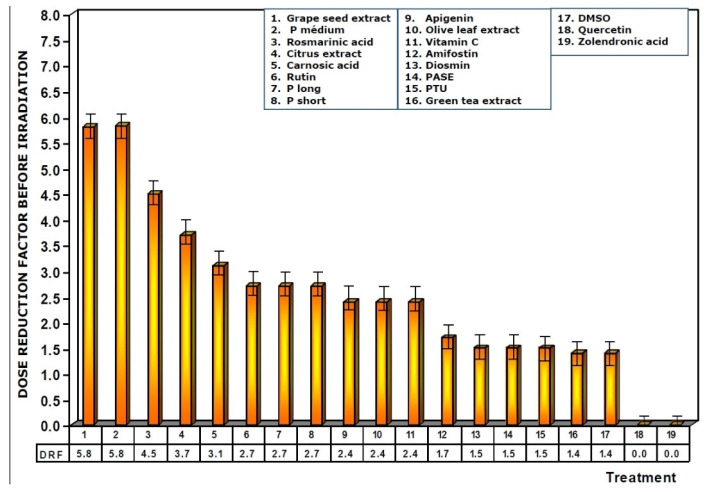
Dose reduction factors (DRF) obtained when the test substances were administered immediately before exposure to ionizing radiation.

**Figure 6 antioxidants-11-00094-f006:**
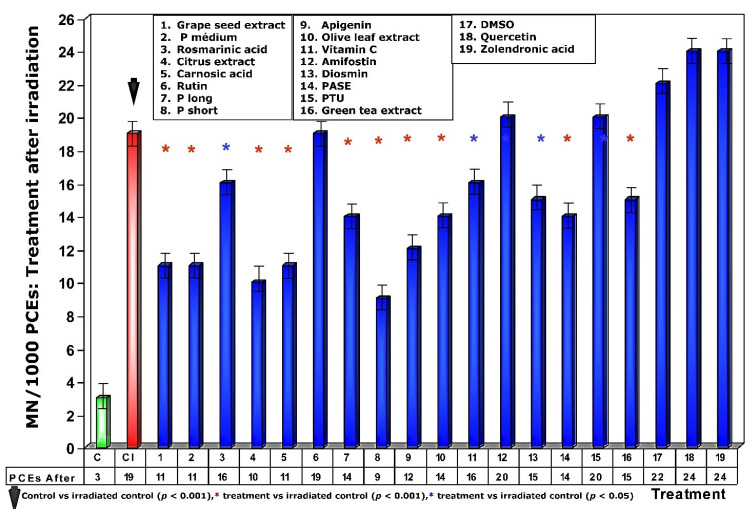
Frequency of micronuclei in polychromatic erythrocytes treated with test substances immediately after exposure to ionizing radiation.

**Figure 7 antioxidants-11-00094-f007:**
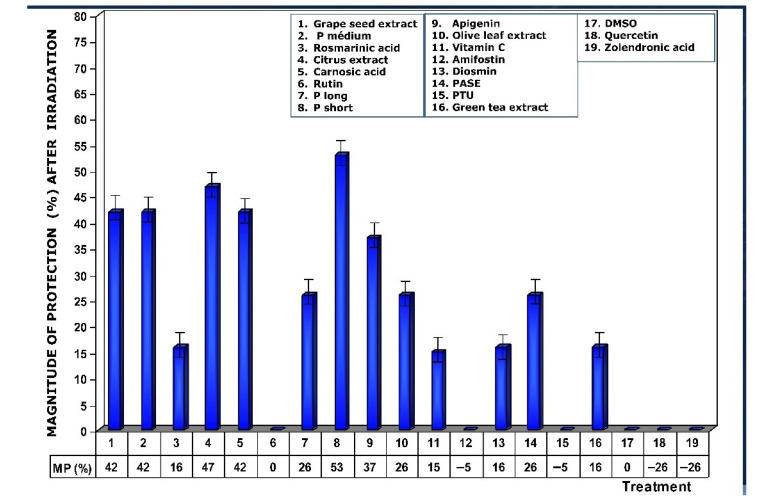
Magnitudes of protection determined when test substances were administered immediately after exposure to ionizing radiation.

**Figure 8 antioxidants-11-00094-f008:**
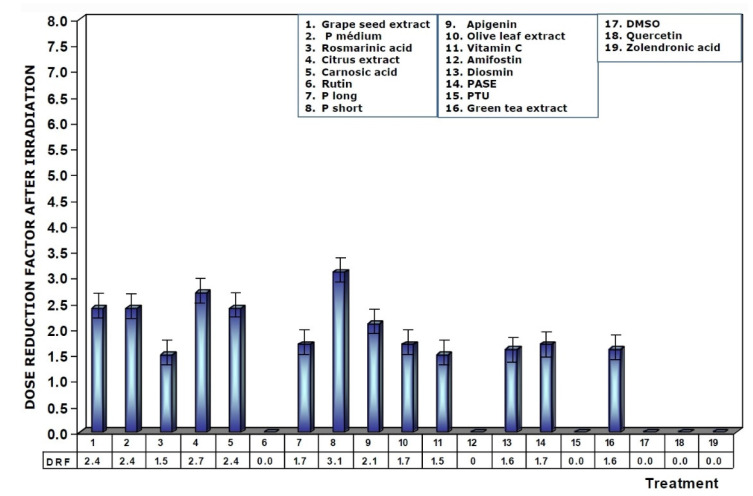
Dose reduction factors (DRFs) obtained when test substances were administered immediately after exposure to ionizing radiation.

**Figure 9 antioxidants-11-00094-f009:**
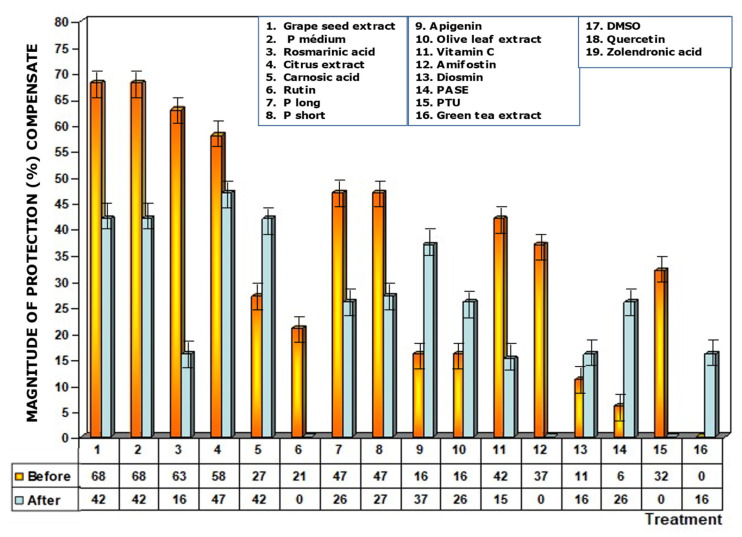
Compensated magnitudes of protection of the substances tested after eliminating possible additive effect of the DMSO used to solubilize the said substances for use in the biological environment. Compensation was carried out on water-insoluble substances that were dissolved in DMSO. Compensated results are shown for substances administered before and immediately after exposure to ionizing radiation.

**Figure 10 antioxidants-11-00094-f010:**
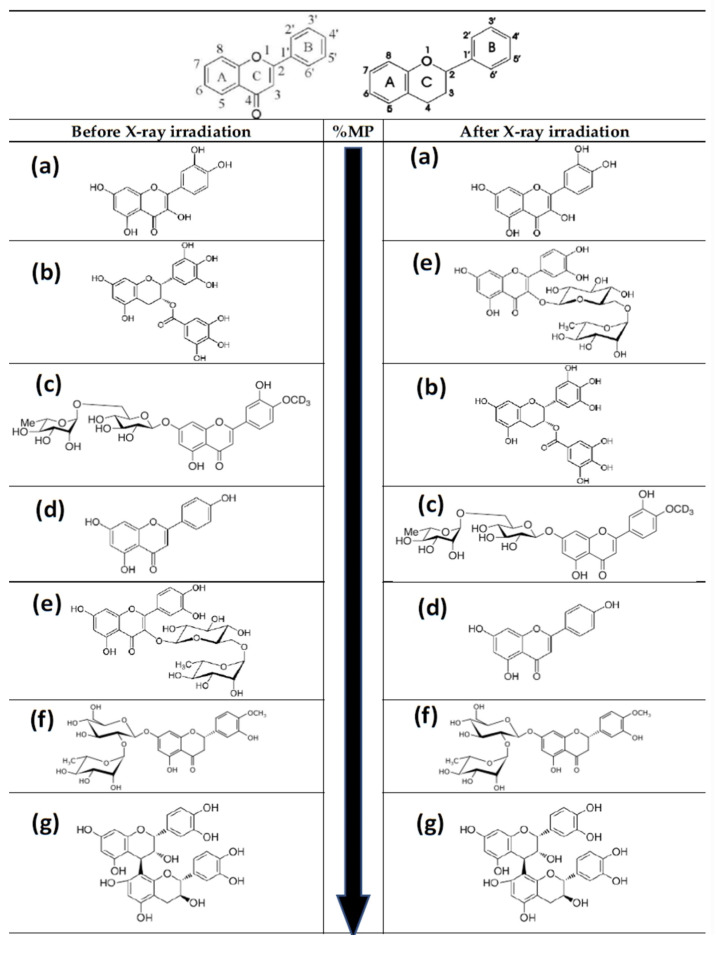
Basic structure of flavonoid skeleton with and without C-4 carbonyl group. Molecular structures of the flavonoids tested in this study from low to high (top to bottom) genoprotective capacity (see Table 2): (**a**) quercetin; (**b**) green tea extract (epigallocatechin-3-O-gallate, EGCG); (**c**) diosmin; (**d**) apigenin; (**e**) rutin; (**f**) citrus extract (neohesperidin); (**g**) procyanidins. Before irradiation: Quercetin < EGCG (green tea) < Diosmin < Apigenin < Rutin < Neohesperidin (citrus extract) < Procyanidins. After irradiation: Quercetin < Rutin < EGCG (green tea) < Diosmin < Apigenin < Neohesperidin < Procyanidins.

**Table 1 antioxidants-11-00094-t001:** Structural characteristics of the tested substances related to the magnitude of protection (%) determined according to the time of their administration in relation to the exposure to X-rays, as well as modifications that could be due to a possible additive effect of the substances dissolved in DMSO.

No	COMPOUND	%MP BEFORE	%MP AFTER	DMSO	%MP BEFORE COMPENSATE	%MP AFTER COMPENSATE *	STRUCTURAL CONSIDERATIONS
1	**Grape seed Ex.**	68	42	N	68	42 (−38%)	Flavonoid: flavan-3-ols (catechins); mixture of monomers, dimers and polymers C1-C15 catechin units. Presence of catechol and gallic groups. Water soluble
2	**P medium**	68	42	N	68	42 (−38%)	Flavonoid: flavan-3-ols (catechins); mixture of monomers, dimers and polymers C1-C15 catechin units. Presence of catechol and gallic groups. Water soluble. 10% of monomers and dimers
3	**Rosmarinic acid**	63	16	N	63	16 (−75%)	Di-Caffeoyl compound; free carboxilic group. Water soluble
4	**Citrus Ex.**	58	47	N	58	47 (−19%)	Flavonoid: flavanone glycosides (naringin and neohesperidin). NO C2 = C3 double bond, NO catechol group. Water soluble.
5	**Carnosic acid**	53	42	Y	27	42 (+56%)	Diterpene: free carboxilic group; catechol group. NO water soluble, lipid soluble.
6	**Rutin**	47	0	Y	21	0 (−100%)	Flavonoid: flavonol glycoside; C2 = C3 double bond; catechol group. NO water soluble.
7	**P long**	47	26	N	47	26 (−45%)	Flavonoid: flavan-3-ols (catechins); mixture of monomers, dimers and polymers C1-C15 catechin units. Presence of catechol and gallic groups. Sligthy watersoluble. 1% of monomers and dimers
8	**P short**	47	53	N	47	53 (+13%)	Flavonoid: flavan-3-ols (catechins); mixture of monomers, dimers and polymers C1-C15 catechin units. Presence of catechol and gallic groups. Watwr soluble,25% of monomers and dimers
9	**Apigenin**	42	37	Y	16	37 (+131%)	Flavonoid: flavone aglycon; C2 = C3 double bond; NO catechol group. NO water soluble.
10	**Olive leaf Ex.**	42	26	Y	16	26 (+63%)	Secoiridoid + minor flavonoids: oleuropein; catechol group and sterified carbonyl groups. Partially water soluble.
11	**Vitamin C**	42	15	N	42	15 (−64%)	(R)-3,4-dihydroxy-5-((S)-1,2-dihydroxyethyl) furan-2(5H)-one. Presence of orto-dihydroxy structure, Water soluble.
12	**Amifostine**	37	−5	N	37	−5 (−114%)	S-phospho derivative of 2-[(3-aminopropyl) amino] ethanethiol. Organothiophosphate with a free amino group. Water soluble
13	**Diosmin**	37	16	Y	11	16 (+45%)	Flavonoid: flavone glycoside; C2 = C3 double bond; NO catechol group. NO water soluble.
14	**PASE**	32	26	Y	6	26 (+333%)	Sargahydroquinoic acid ((2Z,6E,10E)-12-(2,5-dihydroxy-3-methylphenyl)-6,10-dimethyl-2-(4-methylpent-3-enyl) dodeca-2,6,10-trienoic acid, Structure para-hydroxy phenolic and free carboxilic group. NO water soluble.
15	**PTU**	32	−5	N	32	−5 (−116%)	Propylthiouracil. Water soluble (after dissolution in NaOH 0.15N adjusting to pH 8.5).
16	**Green tea Ex.**	26	16	Y	0	16 (>+100%)	Flavonoid: flavan-3-ols (catechins: EGCG, EGC. ECG….all are monomers); presence of catechol and mainly gallic groups. Sligthy watersoluble
17	**DMSO**	26	0	Y			Dimethylsulfoxide. Water soluble
18	**Quercetin**	−16	−26	Y	−42	−26	Flavonoid: flavonol aglycon; C2 = C3 double bond; catechol group, five free hydroxyl groups (can be pro-oxidant). NO water soluble
19	**Zoledronic acid**	−32	−26	N	−58	−26	Imidazole (biphosphonate): 2,2-bis(phosphono)-2-hydroxyethane-1-yl. Sparingly soluble

* % relative modification vs. % MP Before Compensate.

**Table 2 antioxidants-11-00094-t002:** Modifications in the chemical structure of the flavonoids tested from lower to higher genoprotective capacity, including the most significant structures of the extracts used in this study.

Magnitude of Protection (Before X rays)	Magnitude of Protection (After X rays)
Flavonol aglycon (quercetin)	Flavonol aglycon (quercetin)
Flavan-3-ol galloylated (green tea extract)	Flavonol 3-O-glycoside (rutin)
Flavone 7-O-glycoside (diosmin)	Flavan-3-ol galloylated (green tea extract)
Flavone aglycon (apigenin)	Flavone 7-O-glycoside (diosmin)
Flavonol 3-O-glycoside (rutin)	Flavone aglycon (apigenin)
Flavanone 7-O-glycoside (citrus extract)	Flavanone 7-O-glycoside (citrus extract)
Flavan-3-ol (procyanidins)	Flavan-3-ol (procyanidins)

## Data Availability

The data are contained within the article or Appendix A.

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
