# Peer review of "Genoprotective Effect of Some Flavonoids against Genotoxic Damage Induced by X-rays In Vivo: Relationship between Structure and Activity"

_antioxidants, 2021, doi:10.3390/antiox11010094_

Round 1

Reviewer 1 Report

This manuscript is recommended for publication. Fn extensive factual material of great scientific interest which suggest further investigation. For example, the association of antioxidant activity with radioprotective  effect of flavonoids. How identical are pharmacokinetics and dynamics of studied compounds

Reviewer 2 Report

In this manuscript, the authors use a micronucleus assay to characterize different flavonoids for the ability to limit radiation-induced DNA damage in mouse bone marrow cells either before or after irradiation. The then use this information to infer chemical constituents that increase or decrease radioprotection. Given the important role of DNA damage in radiation therapy and the impact of dietary and other flavonoids in this process, this work will be of interest to the relevant fields of study. 

I do not have a major concerns. However, the following minor issues should be addressed to improve the manuscript:

  • The final sentence of the abstract is very long. Consider revising it to make it easier to read.
  • Line 123: remove the word been
  • The authors use cGy in the text but mGy in Figure 1. It would be better to use a single consistent unit.
  • It is not clear to me what is being graphed in Figure 1B. What do the authors mean by expected statistical results? Why can't the linear regression and R squared value be calculated from the actual data in Figure 1A?
  • In Figure 9, the authors should clarify what the orange/yellow and blue bars represent. Presumably, these bars represent before and after irradiation, but this is not define. In addition there appears to be an extra error bar to the left of the y-axis that should be removed.

Reviewer 3 Report

The manuscript by Alcaraz and coworkers describes the genoprotective effect of selected flavonoids against X-ray induced genotoxic damage in vivo. They were intraperitoneally administrated to mice before or immediately after exposure to ionizing radiation. The results provide valuable information to develop radioprotectors based on the natural antioxidative substances. This manuscript is recommended for publication after addressing the below minor issues.

1) Since the mix of cGy and mGy is confusing, cGy should be replaced by mGy.

2) Abbreviations, such as MN (page 5, line 177), DRF (page 5, line 178), and so on, should be defined at their first mention in the text.

3) According to the definitions of the International Union of Pure and Applied Chemistry (IUPAC), the OH group is called “hydroxy group” rather than “hydroxyl group.” The term “hydroxyl” exclusively refers to the hydroxyl radical. So I recommend to replace “hydroxyl group” by “hydroxy group” all over the manuscript.

4) Page 6, line 229: “it should is worthy” should be “it should be worthy.”

5) Page 8, caption of Figure 4: please remove “to” to be “before exposure to…”

6) Page 8, Figure 5: Commas should be replaced by dots for decimal points.

7) Page 9, line 265: “gen-protective” should be “genoprotective.”

8) Page 12, line 379: Sulfhydryl bonds are not SH-SH, but SH. Sulfhydryl group means SH.

9) Page 13, line 434: Charged radical anions, such as superoxide anion, are often indicated with a superscript dot followed by a minus sign.

10) Page 13, line 440: “•HO” should be “•OH.”

11) Page 13, line 446: Flavonoids don’t donate a proton, but hydrogen atom (hydrogen radical, H•) or electron to free radicals. In the latter case, the electron transfer (ET) is often accompanied by proton transfer (PT), i.e., ET followed by PT, or PT followed by ET. For both cases, ET is important to scavenge free radicals.

12) Page 13, line 447: “he” should be “the.”

13) Page 14: The reaction in line 468 is confusing, since OO• means a peroxyl radical. Please use the structural formulae of the catechol.

14) Page 14, line 483: “3,4-dihydroxy” should be “3’,4’-dihydroxy” for the B ring.

15) Page 14, line 504: What is hydrogen peroxide radical? The authors mean hydroperoxyl radical, HOO•?

16) Page 15, line 542: Please add a minus after dot to be O2•-.

17) Page 15, line 546: Please remove “o” after “DMSO.”

18) Page 16, line 570: Do the authors mean “ring B (not ring C)” for the catechol structure? Please confirm it.

19) Page 18, line 614: Please remove “some” after “even when.”

20) Page 18, line 615: Please remove “by.”

21) Page 18, line 626: Please remove “the” between “flavones” and “apigenin.”
